# FEDERATED LEARNING WITH DECOUPLED PROBABILISTIC-WEIGHTED GRADIENT AGGREGATION

## ABSTRACT

In the federated learning paradigm, multiple mobile clients train local models independently based on datasets generated by edge devices, and the server aggregates parameters/gradients from local models to form a global model. However, existing model aggregation approaches suffer from high bias on both data distribution and parameter distribution for non-IID datasets, which result in severe accuracy drop for increasing number of heterogeneous clients. In this paper, we proposed a novel decoupled probabilistic-weighted gradient aggregation approach called *FeDEC* for federated learning. The key idea is to optimize gradient parameters and statistical parameters in a decoupled way, and aggregate the parameters from local models with probabilistic weights to deal with the heterogeneity of clients. Since the overall dataset is unaccessible by the central server, we introduce a variational inference method to derive the optimal probabilistic weights to minimize statistical bias. We further prove the convergence bound of the proposed approach. Extensive experiments using mainstream convolutional neural network models based on three federated datasets show that FeDEC significantly outperforms the state-of-the-arts in terms of model accuracy and training efficiency.

## 1    INTRODUCTION

Federated learning (FL) has emerged as a novel distributed machine learning paradigm that allows a global machine learning model to be trained by multiple mobile clients collaboratively. In such paradigm, mobile clients train local models based on datasets generated by edge devices such as sensors and smartphones, and the server is responsible to aggregate parameters/gradients from local models to form a global model without transferring data to a central server. Federated learning has been drawn much attention in mobile-edge computing (Konecný et al. (2016); Sun et al. (2017)) with its advantages in preserving data privacy (Zhu & Jin (2020); Jiang et al. (2019); Keller et al. (2018)) and enhancing communication efficiency (Shamir et al. (2014); Smith et al. (2018); Zhang et al. (2013); McMahan et al. (2017); Wang et al. (2020)).

Gradient aggregation is the key technology of federated learning, which typically involves the following three steps repeated periodically during training process: (1) the involved clients train the same type of models with their local data independently; (2) when the server sends aggregation signal to the clients, the clients transmit their parameters or gradients to the server; (3) when server receives all parameters or gradients, it applies an aggregation methods to the received parameters or gradients to form the global model. The standard aggregation method FedAvg (McMahan et al. (2017)) and its variants such as FedProx (Li et al. (2020a)), Zeno (Xie et al. (2019)) and q-FedSGD (Li et al. (2020b)) applied the synchronous parameter averaging method to the entire model indiscriminately. Agnostic federated learning (AFL) (Mohri et al. (2019)) defined an agnostic and risk-averse objective to optimize a mixture of the client distributions. FedMA (Wang et al. (2020)) constructed the shared global model in a layer-wise manner by matching and averaging hidden elements with similar feature extraction signatures. The recurrent neural network (RNN) based aggregator (Ji et al. (2019)) learned an aggregation method to make it resilient to Byzantine attack.

Despite the efforts that have been made, applying the existing parameter aggregation methods for large number of heterogeneous clients in federated learning still suffers from performance issues. It was reported in (Zhao et al. (2018)) that the accuracy of a convolutional neural network (CNN) model trained by FedAvg reduces by up to 55% for highly skewed non-IID dataset. The work of

(Wang et al. (2020)) showed that the accuracy of FadAvg (McMahan et al. (2017)) and FedProx (Li et al. (2020a)) dropped from 61% to under 50% when the client number increases from 5 to 20 under heterogeneous data partition. A possible reason to explain the performance drops in federated learning could be the different levels of bias caused by inappropriate gradient aggregation, on which we make the following observations.

**Data Bias:** In the federated learning setting, local datasets are only accessible by the owner and they are typically non-IID. Conventional approaches aggregate gradients uniformly from the clients, which could cause great bias to the real data distribution. Fig. 1 shows the distribution of the real dataset and the distributions of uniformly taking samples from different number of clients in the CIFAR-10 dataset (Krizhevsky (2009)). It is observed that there are great differences between the real data and the sampled distributions. The more clients involved, the more difference occurs.

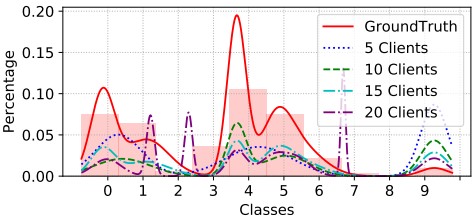 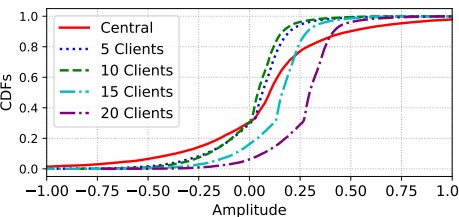

Figure 1: The differences between real data and sampled datasets (CIFAR-10).

Figure 2: The cumulative distribution function of means and variances in BN layers of different FL models (ResNet18@CIFAR-10).

**Parameter Bias:** A CNN model typically contains two different types of parameters: the *gradient parameters* from the convolutional (Conv) layers and full connected (FC) layers; and the *statistical parameters* such as mean and variance from the batch normalization (BN) layers. Existing approaches such as FedAvg average the entire model parameters indiscriminately using distributed stochastic gradient descent (SGD), which will lead to bias on the means and variances in BN layer. Fig. 2 shows the means and variances in BN layer distribution of a centrally-trained CNN model and that of FedAvg-trained models with different number of clients on non-IID local datasets. It is observed that the more clients involved, the larger deviation between the central model and the federated learning models.

**Our contributions**: In the context of federated learning, the problems of data bias and parameter bias have not been carefully addressed in the literature. In this paper, we propose a novel gradient aggregation approach called FeDEC. The main contribution of our work are summarized as follows. (1) We propose the key idea of optimizing gradient aggregation with a decoupled probabilistic-weighted method. To the best of our knowledge, we make the first attempt to aggregate gradient parameters and statistical parameters separatively, and adopt a probabilistic mixture model to resolve the problem of aggregation bias for federated learning with heterogeneous clients. (2) We propose a variational inference method to derive the optimal probabilistic weights for gradient aggregation, and prove the convergence bound of the proposed approach. (3) We conduct extensive experiments using five mainstream CNN models based on three federated datasets under non-IID conditions. It is shown that FeDEC significantly outperforms the state-of-the-arts in terms of model accuracy and training efficiency.

## 2 RELATED WORK

We summarize the related work as two categories: parameter/gradient aggregation for distributed learning and federated learning.

**Distributed Learning**: In distributed learning, the most famous parameter aggregation paradigm is the Parameter Server Framework (Li et al. (2014)). In this framework, multiple servers maintain a partition of the globally shared parameters and communicate with each other to replicate and migrate parameters, while the clients compute gradients locally with a portion of the training data, and communicate with the server for model update. Parameter server paradigm had motivated the development of numerous distributed optimization methods (Boyd et al. (2011); Dean et al. (2012); Dekel et al. (2012); Richtárik & Takác (2016); Zhang et al. (2015)). Several works focused on

improving the communication-efficiency for distributed learning (Shamir et al. (2014); Smith et al. (2018); Zhang et al. (2013)). To address the issue of model robustness, Zeno (Xie et al. (2019)) was proposed to make distributed machine learning tolerant to an arbitrary number of faulty workers. The RNN based aggregator (Ji et al. (2019)) adopted a meta-learning approach that utilizes a recurrent neural network (RNN) in the parameter server to learn to aggregate the gradients from the workers, and designed a coordinatewise preprocessing and postprocessing method to improve its robustness.

**Federated Learning**: Federated learning (Konečný et al. (2015)) is an emerging edge distributed machine learning paradigm that aims to build machine-learning models based on datasets distributing across multiple clients. One of the standard parameter aggregation methods is FedAvg (McMahan et al. (2017)), which combines local stochastic gradient descent (SGD) on each client with a server that performs parameter averaging. The lazily aggregated gradient (Lag) method (Chen et al. (2018)) allowed clients running multiple epochs before model aggregation to reduce communication cost. For heterogeneous datasets, FedProx (Li et al. (2020a)) modified FedAvg by adding a heterogeneity bound on datasets and devices to tackle heterogeneity. The FedMA (Wang et al. (2020)) method, derived from AFL Mohri et al. (2019) and PFNM (Yurochkin et al. (2019)), demonstrated that permutations of layers can affect the gradient aggregation results, and proposed a layer-wise gradient aggregation method to solve the problem. For fair resources allocation, the q-FedSGD (Li et al. (2020b)) method encouraged a more uniform accuracy distribution across devices in federated networks.

However, all the methods did not differentiate gradient parameters and statistical parameters and aggregated the entire model in a coupled manner. In this paper, we make the first attempt to decouple the aggregation of gradient parameters and statistical parameters with probabilistic weights to optimize the global model to achieve fast convergence and high accuracy in non-IID conditions.

## 3 FEDEC: A DECOUPLED GRADIENT AGGREGATION METHOD

### 3.1 OBJECTIVE OF FEDERATED LEARNING WITH NON-IID DATA

Consider a federated learning scenario with $K$ clients that train their local CNN models independently based on local datasets $\mathbf{x}_1, \mathbf{x}_2, \ldots, \mathbf{x}_K$ and report their gradients and model parameters to a central server. The objective of the server is to form an aggregate global CNN model to minimize the loss function over the total datasets $\mathbf{x} = \{\mathbf{x}_1, \mathbf{x}_2, \ldots, \mathbf{x}_K\}$. Conventional federated learning tends to optimize the following loss function:

$$\min_{\mathbf{W}} \mathcal{L}(\mathbf{W}, \mathbf{x}) := \sum_{k=1}^{K} \frac{|\mathbf{x}_k|}{|\mathbf{x}|} \mathcal{L}_k(\mathbf{W}_k, \mathbf{x}_k), \tag{1}$$

where $\mathbf{W}$ is the parameters of the global model, $\mathbf{W}_k$ $(k = 1, 2, \cdots, K)$ is the parameters of the $k$-th local model; $\mathcal{L}(\cdot)$ and $\mathcal{L}_k(\cdot)$ indicate the loss functions for global model and local models accordingly. The above objective assumes training samples uniformly distributed among the clients, so that the aggregated loss can be represented by the sum of percentage-weighted of the local losses.

As discussed in section 1, conventional federated learning has two drawbacks. Firstly, local datasets are collected by mobile devices used by particular users, which are typically non-IID. Training samples on each client may be drawn from a different distribution, therefore the data points available locally could be bias from the overall distribution. Secondly, since a neural network model is typically consists of convolutional (Conv) layers and full-connected (FC) layers that are formed by gradient parameters, and batch normalization (BN) layers that are formed by statistical parameters such as mean and variance, aggregating them without distinction will cause severe deviation of the global model parameters.

To address the above issues, we propose a decoupled probabilistic-weighted approach for federated learning that focuses on optimizing the following loss function:

$$\min_{\mathbf{W}_*} \mathcal{L}(\{\mathbf{W}_{NN}^t, \mathbf{W}_{mean}^t, \mathbf{W}_{var}^t\}, \mathbf{x}) := \sum_{k=1}^{K} \pi_k \mathcal{L}_k(\{\mathbf{W}_{NN}^{t-1,k}, \mathbf{W}_{mean}^{t-1,k}, \mathbf{W}_{var}^{t-1,k}\}, \mathbf{x}_k), \tag{2}$$

where $*$ indicates $NN, mean$ and $var$; $\mathbf{W}_{NN}^t$, $\mathbf{W}t_{mean}$ and $\mathbf{W}_{var}^t$ are the parameters of Conv and FC layers of the global model after $t$-th aggregation epoch; $\mathbf{W}_{NN}^{t-1,k}$, $\mathbf{W}_{mean}^{t-1,k}$ and $\mathbf{W}_{var}^{t-1,k}$ are the $k$-th local model been trained several local epoch based $t-1$-th global model; $\pi_k (k = 1, \ldots, K)$ is

the probability that a sample is drawn from the distribution of the $k$-th client, i.e., $\pi_k \in [0,1]$ ($k = 1, \ldots, K$) and $\sum_{k=1}^{K} \pi_k = 1$.

The above formulation objects to minimize the expected loss over K clients with non-IID datasets. Next, we will introduce a decoupled method called *FeDEC* to optimize the parameters of different types of layers separatively, and derived the probability weights $\pi_k$ for parameter aggregation.

## 3.2 DECOUPLED PROBABILISTIC-WEIGHTED GRADIENT AGGREGATION METHOD

In this section, we proposed a decoupled method to derive the global model with respect to $\mathbf{W}_{NN}^t$ (parameters of Conv and FC layers) and $\mathbf{W}_{mean}^t, \mathbf{W}_{var}^t$ (statistical parameters of BN layers).

### 3.2.1 GRADIENT AGGREGATION FOR CONV AND FC LAYERS

Since the parameters of Conv and FC layers are neural network weights which are updated by distributed gradient descent method (Nesterov (1983)), they are appropriate to be aggregated with a similar approach that adapts conventional federated average for non-IID datasets. Let $\mathbf{g}_k^t = \mathbf{W}_{NN*}^{t-1,k} - \mathbf{W}_{NN}^{t-1,k}$ ($k = 1, \ldots, K$), where $NN*$ indicates $NN$ parameters after full local training. be the gradient of the $k$-th client in the $t$-th training epoch. After receiving the gradients from $K$ clients, the central server update the parameters of global model as follows.

$$\mathbf{W}_{NN}^t = \mathbf{W}_{NN}^{t-1} - \beta \sum_{k=1}^{K} \pi_k^t \mathbf{g}_k^t, \tag{3}$$

where $\beta$ is the learning rate for parameter update, $\pi_k^t$ ($k = 1, \ldots, K$) are the probabilistic weights with $\sum_{k=1}^{K} \pi_k^t = 1$ that are derived in section 3.2.3.

### 3.2.2 PARAMETER AGGREGATION FOR MEANS AND VARIANCES IN BN

Different from the Conv and FC layers, the BN layers mainly contain statistical parameters such as mean and variance. Conventional federated learning aggregates BN layers and other layers without distinction, which could lead to high bias of means and variances in BN layer of the global model. Thus we propose a different way to aggregate means and variances in BN layer as follows.

In $t$-th training epoch, the means and variances $\mathbf{W}_{mean}^t, \mathbf{W}_{var}^t$ in BN layer, which are updated by:

$$\mathbf{W}_{mean}^t = \sum_{k=1}^{K} \pi_k^t \mathbf{W}_{mean}^{t,k}, \tag{4}$$

$$\mathbf{W}_{var}^t = \frac{1}{|\mathbf{x}| - K} \sum_{k=1}^{K} (|\mathbf{x}_k| - 1)\pi_k^t \mathbf{W}_{var}^{t,k}, \tag{5}$$

where $\mathbf{W}_{mean}^{t,k}$ and $\mathbf{W}_{var}^{t,k}$ indicate the means and variances in BN layers of the $k$-th client in epoch $t$; $\pi_k^t$ ($k = 1, \ldots, K$) are probabilistic weights with $\sum_{k=1}^{K} \pi_k^t = 1$ that are derived in section 3.2.3.

In the above equations, we update the mean with the weighted average of local models, and update the variance with the weighted pooled variance (Killeen (2005)), which can give an unbias estimation of parameters of the whole dataset under non-IID conditions (see Appendix A.2).

### 3.2.3 DERIVATION OF PROBABILISTIC WEIGHTS

We adopt a mixture probabilistic model to describe non-IID datasets in federated learning. Without loss of generality, in the $t$-th training epoch, we assume the mini-batch samples of each client follows a Gaussian distribution $\mathcal{N}_k(\mu_k, \sigma_k)$ ($k = 1, \ldots, K$), where $\mu_k, \sigma_k$ are the mean and standard deviation of the distribution that vary among clients. We omit the upper script $t$ for simplicity thereafter. The whole samples can be described as a Gaussian Mixture Model (GMM) with the following probability function:

$$p(\mathbf{x}|\lambda) = \sum_{k=1}^{K} \pi_k p(\mathbf{x}_k|\mu_k, \sigma_k), \tag{6}$$

where $\lambda = \{\pi_k, \mu_k, \sigma_k \mid k = 1, 2, \cdots, K\}$ are the parameters of the GMM model[1].

---

[1]Noted that the proposed variational inference method can be applied to other non-Gaussian distributions with slight modification.

In federated learning, the local data samples are accessed by particular client and the central server can only observe the statistics of local dataset such as mean and standard variance. Without knowing the overall samples, conventional expectation-maximization (EM) algorithm (Dempster et al. (1977)) cannot be applied to derive $\lambda$. Alternatively, we introduce a variational inference method to estimate the parameters of $\lambda$.

Specifically, we construct a variational Bayesian generative model to generate data that are close to the reported statistics of local models as possible, and use the generated data to estimate the GMM model parameters. The plate notions of the generative model are shown in Fig. 3. The notations are explained as follows.

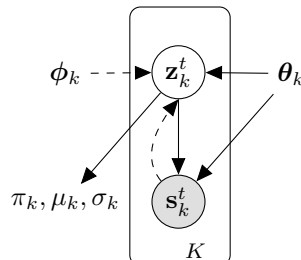

• $\mathbf{s}_k^t = \{\mathbf{W}_{mean}^{t,k}, \mathbf{W}_{var}^{t,k}\}$ is the observed statistics from the feature maps of $k$-th client.

• $\mathbf{z}_k^t = \{\mathbf{z}_{k,i}^t | (i = 1, 2, \cdots, C)\}$ is a vector of latent variables with length $C$, where $\mathbf{z}_{k,i}^t \in [0, 1]$, $\sum_{i=1}^{C} \mathbf{z}_{k,i}^t = 1$, and $C$ is the number of classes for a classification task. $\mathbf{z}_k^t$ can be viewed as a data distribution that represents the probability of a sample in client $k$ belonging to the classes.

Figure 3: The variational Bayesian generative model using plate notations.

• $\boldsymbol{\theta} = \{\boldsymbol{\theta}_k\}$ are generative model parameters, and $\boldsymbol{\phi} = \{\boldsymbol{\phi}_k\}$ are variational parameters.

The solid lines in Fig. 3 denote the generative model $p_{\boldsymbol{\theta}_k}(\mathbf{z}_k^t)p_{\boldsymbol{\theta}_k}(\mathbf{s}_k^t|\mathbf{z}_k^t)$, and the dashed lines denote the variational approximation $q_{\boldsymbol{\phi}_k}(\mathbf{z}_k^t|\mathbf{s}_k^t)$ to the intractable posterior $p_{\boldsymbol{\theta}_k}(\mathbf{z}_k^t|\mathbf{s}_k^t)$ We approximate $p_{\boldsymbol{\theta}_k}(\mathbf{z}_k^t|\mathbf{s}_k^t)$ with $q_{\boldsymbol{\phi}_k}(\mathbf{z}_k^t|\mathbf{s}_k^t)$ by minimizing their divergence:

$$\boldsymbol{\phi}_k^*, \boldsymbol{\theta}_k^* = arg \min_{\boldsymbol{\theta}_k, \boldsymbol{\phi}_k} divergence(q_{\boldsymbol{\phi}_k}(\mathbf{z}_k^t|\mathbf{s}_k^t) \,||\, p_{\boldsymbol{\theta}_k}(\mathbf{z}_k^t|\mathbf{s}_k^t)),$$

$$s.t. \sum_{i=1}^{C} \mathbf{z}_{k,i}^t = 1. \tag{7}$$

To derive the optimal value of the parameters $\boldsymbol{\phi}_k$ and $\boldsymbol{\theta}_k$, we compute the marginal likelihood of $\mathbf{s}_k^t$:

$$\log p(\mathbf{s}_k^t) = D_{KL}(q_{\boldsymbol{\phi}_k}(\mathbf{z}_k^t|\mathbf{s}_k^t) \,||\, p_{\boldsymbol{\theta}_k}(\mathbf{z}_k^t|\mathbf{s}_k^t)) + \mathbb{E}_{q_{\boldsymbol{\phi}_k}(\mathbf{z}_k^t|\mathbf{s}_k^t)}\left[\log \frac{p_{\boldsymbol{\theta}_k}(\mathbf{z}_k^t, \mathbf{s}_k^t)}{q_{\boldsymbol{\phi}_k}(\mathbf{z}_k^t|\mathbf{s}_k^t)}\right]. \tag{8}$$

In Eq. 8, the first term is the KL-divergence (Joyce (2011)) of the approximate distribution and the posterior distribution; the second term is called the ELBO (Evidence Lower BOund) on the marginal likelihood of dataset in the $k$-th client.

Since $\log p(\mathbf{s}_k^t)$ is non-negative, the minimization problem of Eq. 7 can be converted to maximize the ELBO. To solve the problem, we change the form of ELBO as:

$$\mathbb{E}_{q_{\boldsymbol{\phi}_k}(\mathbf{z}_k^t|\mathbf{s}_k^t)}\left[\log \frac{p_{\boldsymbol{\theta}_k}(\mathbf{z}_k^t, \mathbf{s}_k^t)}{q_{\boldsymbol{\phi}_k}(\mathbf{z}_k^t|\mathbf{s}_k^t)}\right] = \underbrace{\mathbb{E}_{q_{\boldsymbol{\phi}_k}(\mathbf{z}_k^t|\mathbf{s}_k^t)}\left[log \frac{p(\mathbf{z}_k^t)}{q_{\boldsymbol{\phi}_k}(\mathbf{z}_k^t|\mathbf{s}_k^t)}\right]}_{\text{Encoder}} + \underbrace{\mathbb{E}_{q_{\boldsymbol{\phi}_k}(\mathbf{z}_k^t|\mathbf{s}_k^t)}[\log p_{\boldsymbol{\theta}_k}(\mathbf{s}_k^t|\mathbf{z}_k^t)]}_{\text{Decoder}}. $$

$$\tag{9}$$

The above form is a variational encoder-decoder structure: the model $q_{\boldsymbol{\phi}_k}(\mathbf{z}_k^t|\mathbf{s}_k^t)$ can be viewed as a probabilistic encoder that given an observed statistics $\mathbf{s}_k^t$ it produces a distribution over the possible values of the latent variables $\mathbf{z}_k^t$; The model $p_{\boldsymbol{\theta}_k}(\mathbf{s}_k^t|\mathbf{z}_k^t)$ can be refered to as a probabilistic decoder that reconstructs the value of $\mathbf{s}_k^t$ based on the code $\mathbf{z}_k^t$. According to the theory of variational inference (Kingma & Welling (2014)), the problem in Eq. 9 can be solved with stochastic gradient descent (SGD) method using a fully-connected neural network to optimize the mean squared error loss function.

With the derived optimal parameters $\boldsymbol{\phi}_k^*, \boldsymbol{\theta}_k^*$, we can extract the latent variables $\mathbf{z}_k^t$ that is interpreted as the sample distribution of client-$k$. Therefore $\mathbf{z}_k^t$ can be used to infer the parameters $(\pi_k, \mu_k, \sigma_k)$ of $k$-th component of the GMM model. Specifically, the probabilistic weights $\pi_k$ can be represented by

$$\pi_k^t = \left\{\sum_{i=1}^{C} \frac{\mathbf{z}_{k,i}^t}{\sum_{j=1}^{K} \mathbf{z}_{j,i}^t}\right\} \bigg/ \left\{\sum_{k=1}^{K} \sum_{i=1}^{C} \frac{\mathbf{z}_{k,i}^t}{\sum_{j=1}^{K} \mathbf{z}_{j,i}^t}\right\}. \tag{10}$$

## 4 CONVERGENCE ANALYSIS

In this section, we will show that the convergence of the proposed FeDEC algorithm is theoretically guaranteed. We use the following assumptions and lemmas, and the convergence guarantee is provided in Theorem 1.

**Assumption 1** *(Unbiased Gradient): We assume that the stochastic gradients $\mathbf{g}_i^t$ is an unbiased estimator of the true gradient $\nabla f(\mathbf{w}_i^t)$, i.e., $\mathbb{E}[\mathbf{g}_i^t] = \nabla f(\mathbf{w}_i^t)$, where $f(\cdot)$ is any convex objective function and $\mathbf{w}_i^t$ is its variables.*

**Assumption 2** *(Gradient Convex Set): We assume that gradient set $\mathbf{G}$ is a convex set, where all gradients $\mathbf{g}_1, \mathbf{g}_2, \ldots, \mathbf{g}_K$ are in $\mathbf{G}$, and any $\mathbf{g} = \sum_{i=1}^{K} \lambda_i \mathbf{g}_i$ ($\forall \lambda_i > 0$ and $\sum_{i=1}^{K} \lambda_i = 1$) is in $\mathbf{G}$.*

**Lemma 1** *(L-Lipschitz Continuity): For a function $f(\cdot)$ is Lipschitz continuous if there exists a positive real constant $L$ such that, for all real $x_1$ and $x_2$:*

$$|f(x_1) - f(x_2)| \leq L|x_1 - x_2|.$$

**Lemma 2** *(Jensen's Inequality): If $f(\mathbf{w})$ is a convex function on $\mathcal{W}$, and $\mathbb{E}[f(\mathbf{w})]$ and $f(\mathbb{E}[\mathbf{w}])$ are finite, then:*

$$\mathbb{E}[f(\mathbf{w})] \geq f(\mathbb{E}[\mathbf{w}])).$$

**Definition 1** *(Projection Operation): Assume $\mathbf{w}_*$ is an intermediate result of optimization, we define a project operator $\prod_{\mathcal{W}}(\mathbf{w}_*)$ to project $\mathbf{w}_*$ to the domain $\mathcal{W}$, which is computed by:*

$$\prod_{\mathcal{W}}(\mathbf{w}_*) = arg \min_{\mathbf{w} \in \mathcal{W}} ||\mathbf{w} - \mathbf{w}_*||.$$

**Definition 2** *(Diameter of Domain): Given a function $f(\mathbf{w})$, where $\mathbf{w} \in \mathcal{W}$, and $\mathcal{W}$ is $f$'s domain of definition. The diameter of $\mathcal{W}$ is denoted by $\Gamma$: for every $\mathbf{w}_1, \mathbf{w}_2 \in \mathcal{W}$: $||\mathbf{w}_1 - \mathbf{w}_2|| \leq \Gamma$.*

**Theorem 1** *(Guaranteed Convergence Rate): If a convex function $f(\mathbf{w})$ is L-Lipschitz continuous function, then $||\nabla f(\mathbf{w})|| \leq L$. Let $\Gamma$ be the diameter of domain. Applying equations (3)(4)(5) for gradients aggregation, we have the following convergence rate for the proposed FeDEC algorithm:*

$$f(\bar{\mathbf{w}}^T) - \min_{\mathbf{w} \in \mathcal{W}} f(\mathbf{w}) \leq \mathcal{O}(\frac{\Gamma^2}{2\beta T} + \frac{\beta}{2} L^2), \tag{11}$$

*where $\bar{\mathbf{w}}^T$ is the average result of $\mathbf{w}$ for total training epoch $T$, $\beta$ is the learning rate in equation-(3), and $T$ is the total training epoch. If we let $\beta = \frac{\Gamma}{L\sqrt{T}}$, the convergence rate is $\mathcal{O}(\frac{1}{\sqrt{T}})$.*

**Proof skeleton**: We provide a simple description of the proof skeleton of Theorem 1 with the following steps. (1) Since $f(\cdot)$ is a convex function, we have $f(\mathbf{w}^t) - f(\mathbf{w}) \leq \nabla f(\mathbf{w}^t)(\mathbf{w}^t - \mathbf{w})$. (2) With assumption 1 and 2, we have $f(\mathbf{w}^t) - f(\mathbf{w}) \leq \frac{1}{2\beta}(||\mathbf{w}^t - \mathbf{w}||^2 - ||\mathbf{w}_*^{t+1} - \mathbf{w}||^2) + \frac{\beta}{2}||\nabla f(\mathbf{w}^t)||^2$, where $\mathbf{w}_*^{t+1}$ is the intermediate result of $f(\mathbf{w})$ in update time $t+1$. (3) With lemma 1 and definition 1, by projecting $\mathbf{w}_*^{t+1}$ to $\mathbf{w}^{t+1}$, we have $f(\mathbf{w}^t) - f(\mathbf{w}) \leq \frac{1}{2\beta}(||\mathbf{w}^t - \mathbf{w}||^2 - ||\mathbf{w}^{t+1} - \mathbf{w}||^2) + \frac{\beta}{2}L^2$. (4) Summing from $t = 1$ to $T$ and with definition 1 and 2, we have $\sum_{t=1}^{T} f(\mathbf{w}^t) - Tf(\mathbf{w}) \leq \frac{1}{2\beta}\Gamma^2 + \frac{\beta}{2}L^2 T$. (5) According to lemma 2, we have $f(\bar{\mathbf{w}}^T) - f(\mathbf{w}) \leq \frac{\Gamma^2}{2\beta T} + \frac{\beta}{2}L^2$. (6) Taking $\beta = \Gamma/(L\sqrt{T})$, we can obtain the convergence rate in the theorem. The detailed proof of Theorem 1 and explanations are provided in Appendix A.1.

According to Theorem 1, the FeDEC parameter aggregation algorithm is guaranteed to converge, and the convergence rate can be as fast as general stochastic gradient decent which only related to the training epoch $T$ with an associated constants. The constant is related to the optimization problem parameters such as lipschitz constant $L$, and diameter of domain $\Gamma$.

## 5 PERFORMANCE EVALUATION

In this section, we evaluate the performance of the proposed FeDEC method for federated learning.

## 5.1 EXPERIMENTAL SETUP

**Implementation.** We implement the proposed FeDEC parameter aggregation approach and the considered baselines in PyTorch (Paszke et al. (2019)). We train the models in a simulated federated learning environment consisting of one server and a set of mobile clients with wireless network connections. Unless explicitly specified, the default number of clients is 20, and the learning rate $\beta = 0.01$. We conduct experiments on a GPU-equiped personal computer (CPU: Inter Core i7-8700 3.2GHz, GPU: Nvidia GeForce RTX 2070, Memory: 32GB DDR4 2666MHz, and OS: 64-bit Ubuntu 16.04).

**Models and datasets.** We conduct experiments based on 5 mainstream neural network models: ResNet18 (He et al. (2016)), LeNet (Lecun et al. (1998)), DenseNet121 (Huang et al. (2017)), MobileNetV2 (Sandler et al. (2018)), and a 4-layer CNN (every CNN layer is followed by a BN layer). The detailed structure of the CNN models are provided in Appendix A.3.

We use three real world datasets: MNIST (LeCun et al. (2010)), Fashion-MNIST (Xiao et al. (2017)), and CIFAR-10 (Krizhevsky (2009)). MNIST is a dataset for hand written digits classification with 60000 samples and each example is a $28 \times 28$ greyscale image. Fashion-MNIST is a dataset intended to replace the original MNIST for benchmarking machine learning algorithms. CIFAR-10 is a larger dataset with 10 categories. Each category has 5000 training images and 1000 validation images of size $32 \times 32$. For each dataset, we use 80% of the data for training and amalgamate the remaining data into a global test set.

We form non-IID local datasets as follows. Assume there are $C$ classes of samples in a dataset. Each client draw samples form the dataset with probability $pr(x) = \begin{cases} \eta \in [0, 1], & \text{if } x \in class_j, \\ \mathcal{N}(0.5, 1), & \text{otherwise.} \end{cases}$

It means that the client draw samples from a particular class $j$ with a fixed probability $\eta$, and from other classes based on standard Gaussian distribution. The larger $\eta$ is, the more likely the client's samples concentrate on a particular class, and the more heterogeneous the local datasets are.

## 5.2 PERFORMANCE COMPARISON

We compare the performance of FeDEC with 5 state-of-the-art methods: FedAvg (McMahan et al. (2017)), RNN based aggregator (Ji et al. (2019)), FedProx (Li et al. (2020a)), q-FedSGD (Li et al. (2020b)), and FedMA (Wang et al. (2020)). The results are analyzed as follows.

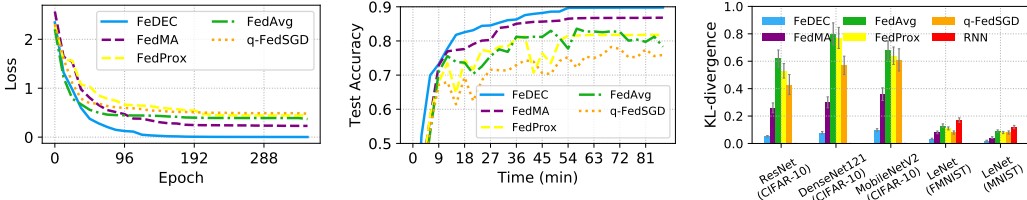

Figure 4: Convergence of different algorithms (ResNet18 on CIFAR-10).

Figure 5: Training accuracy of different algorithms (ResNet18 on CIFAR-10).

Figure 6: KL-divergence of different algorithms.

**Convergence**: In this experiment we study the convergence of all baselines and our algorithm by showing the total communication epochs versus train loss. Fig. 4 shows the result of ResNet18 on CIFAR-10. It is shown that the loss of all algorithms tend to be stable after a number of epochs. Clearly FeDEC has the lowest loss among all algorithms, which means that FeDEC converges faster that of baselines. The results of more CNN models on different datasets are shown in Appendix A.4.

**Training Efficiency**: In this experiment we study the test accuracy versus time during training of a CNN model with federated learning. Fig.5 shown the results of training ResNet18 on CIFAR-10. It is shown that FeDEC reaches $0.8$ accuracy after 18 minutes, while FedMA, FedProx, and FedAvg take 36 to 63 minutes to reach the same accuracy. FeDEC approaches $0.9$ accuracy after 54 minutes, while the accuracy of other algorithms are below $0.85$. The results of more CNN models on different datasets are shown in Appendix A.5. It suggests that FeDEC trains much faster than the baseline algorithms and it can reach high accuracy in a short time period.

**Parameter Bias**: In this experiment we study the parameter bias of federated learning algorithms. Fig. 6 compares the KL-divergence between the means and variances in BN of global models aggregated by different algorithms and the central model. It is shown that FedAvg, FedProx, and q-FedSGD have exceptional high parameter bias, while FeDEC has significantly lower KL-divergence compared to the baselines for different CNN models on different datasets.

**Global Model Accuracy**: In this experiment, we compare the global model accuracy of different federated parameter aggregation algorithms after training to converge. We repeat the experiment for 20 rounds and show the average results in Table 1. As shown in the table, the central method yields the highest accuracy. In comparison of different federated learning methods, FeDEC significantly outperforms the other algorithms in global model accuracy. It performs better than the state-of-the-art method FedMA with 2.87%, 3.17%, 2.58%, and 3.09% accuracy improvement in ResNet18, DenseNet121, MobileNetV2, and 4-L CNN respectively for CIFAR-10, 1.09% improvement in LeNet for F-MNIST, and 0.33% improvement in LeNet for MNIST accordingly. FeDEC achieves the highest accuracy among all baselines, and it performs very close to the centralized method, whose accuracy drop is less than 3% in all cases.

Table 1: Average test accuracy on non-IID datasets. The "**Central**" method trains the CNN model in the central server with global dataset. The "**FeDEC**(w/o)" method means using the proposed probabilistic-weighted aggregation method without distinguishing $NN$ and $mean, var$. The "**FeDEC**" method represents the proposed decoupled probabilistic-weighted aggregation approach.

| Model@ Dataset | Central | FedAvg | RNN | FedProx | q-FedSGD | FedMA | FeDEC(w/o) | FeDEC |
|---|---|---|---|---|---|---|---|---|
| ResNet18@CIFAR-10 | **92.33%** | 83.29% | - | 83.47% | 81.08% | 87.44% | 88.30% | **90.31%** |
| DenseNet121@CIFAR-10 | **93.24%** | 82.36% | - | 85.03% | 82.43% | 88.12% | 89.25% | **91.29%** |
| MobileNetV2@CIFAR-10 | **92.51%** | 83.11% | - | 80.68% | 79.82% | 86.59% | 87.93% | **89.17%** |
| 4-L CNN@CIFAR-10 | **85.53%** | 79.33% | - | 80.02% | 77.77% | 81.76% | 83.11% | **84.85%** |
| LeNet@F-MNIST | **90.42%** | 87.41% | 82.32% | 88.33% | 86.21% | 89.02% | 89.52% | **90.11%** |
| LeNet@MNIST | **98.95%** | 97.32% | 97.06% | 97.55% | 95.88% | 98.16% | 98.37% | **98.49%** |
| BiLSTM@Sent140 | **81.47%** | 72.14% | - | 71.08% | 68.44% | 73.81% | **77.51%** | **77.51%** |

**Hyperparameter Analysis**: We further analyze the influence of two hyperparameters in federated learning: the number of clients involved and the heterogeneity of local datasets.

Fig. 7 compares the test accuracy of the global model for different number of involved clients. According to the figure, the performance of FeDEC is stable. When the number of mobile clients increases from 5 to 20, the test accuracy slightly decreases from 0.909 to 0.893. Other baseline algorithms yield significant performance drop. FeDEC achieves the highest test accuracy among all federated learning algorithms in all cases, and it performs very close to the central model.

In the experiment, the heterogeneity of local datasets is represented by $\eta$, the probability that a client tends to sample from a particular class. The more $\eta$ approaches to 1, the more heterogeneous the local datasets are. Fig. 8 shows the test accuracy under different level of heterogeneity. As $\eta$ increases, the test accuracy of all models decreases. FeDEC yields the highest test accuracy among all algorithms, and its performance drops much slower than the baselines. It verifies the effectiveness of the proposed probabilistic-weighted gradient aggregation approach under non-IID conditions.

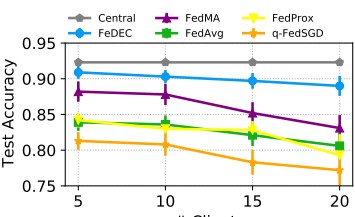

Figure 7: Test accuracy with different number of clients (ResNet18 on CIFAR-10).

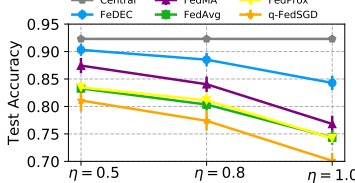

Figure 8: Test accuracy with different level of heterogeneity (ResNet18 on CIFAR-10).

## 6 CONCLUSION

Gradient aggregation played an important role in federated learning to form a global model. To address the problem of data and parameter bias in federated learning for non-IID dataset, we proposed a novel probabilistic parameter aggregation method called FeDEC that decoupled gradient parameters and statistical parameters to aggregate them separatively. The probabilistic weights were optimized with variational inference, and the proposed method was proved to be convergence guaranteed. Extensive experiments showed that FeDEC significantly outperforms the state-of-the-arts on a variety of performance metrics.

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

# A APPENDIX

## A.1 PROOF OF CONVERGENCE GUARANTEE (THEOREM 1 IN SECTION 4)

We provide the detailed proof of Theorem 1 in Section 4. We first restate the necessary equations and the theorem.

In section 3, we propose the usage of the following equations for gradient decent update algorithm:

$$\mathbf{W}_{NN}^t = \mathbf{W}_{NN}^{t-1} - \beta \sum_{k=1}^{K} \pi_k^t \mathbf{g}_k^t, \tag{3}$$

where $\beta$ is the learning rate for parameter update. And for batch normalization update:

$$\mathbf{W}_{mean}^t = \sum_{k=1}^{K} \pi_k^t \mathbf{W}_{mean}^{t,k} \tag{4}$$

$$\mathbf{W}_{var}^t = \frac{1}{||\mathbf{X}|| - n} \sum_{k=1}^{K} (||\mathbf{x}_k|| - 1) \pi_k^t \mathbf{W}_{var}^{t,k} \tag{5}$$

We restate the theorem in section 4 in the following:

**Assumption 1** *(Unbiased Gradient): We assume that the stochastic gradients $\mathbf{g}_i^t$ is an unbiased estimator of the true gradient $\nabla f(\mathbf{w}_i^t)$, i.e., $\mathbb{E}[\mathbf{g}_i^t] = \nabla f(\mathbf{w}_i^t)$, where $f(\cdot)$ is any convex objective function and $\mathbf{w}_i^t$ is its variables.*

**Assumption 2** *(Gradient Convex Set): We assume that gradient set $\mathbf{G}$ is a convex set, where all gradients $\mathbf{g}_1, \mathbf{g}_2, \ldots, \mathbf{g}_K$ are in $\mathbf{G}$, and any $\mathbf{g} = \sum_{i=1}^{K} \lambda_i \mathbf{g}_i$ ($\forall \lambda_i > 0$ and $\sum_{i=1}^{K} \lambda_i = 1$) is in $\mathbf{G}$.*

**Lemma 1** *(L-Lipschitz Continuity): For a function $f(\cdot)$ is Lipschitz continuous if there exists a positive real constant L such that, for all real $x_1$ and $x_2$:*

$$|f(x_1) - f(x_2)| \le L|x_1 - x_2|.$$

**Lemma 2** *(Jensen's Inequality): If $f(\mathbf{w})$ is a convex function on $\mathcal{W}$, and $\mathbb{E}[f(\mathbf{w})]$ and $f(\mathbb{E}[\mathbf{w}])$ are finite, then:*

$$\mathbb{E}[f(\mathbf{w})] \ge f(\mathbb{E}[\mathbf{w}])).$$

**Definition 1** *(Projection Operation): Assume $\mathbf{w}_*$ is an intermediate result of optimization, we define a project operator $\prod_{\mathcal{W}}(\mathbf{w}_*)$ to project $\mathbf{w}_*$ to the domain $\mathcal{W}$, which is computed by:*

$$\prod_{\mathcal{W}}(\mathbf{w}_*) = arg \min_{\mathbf{w} \in \mathcal{W}} ||\mathbf{w} - \mathbf{w}_*||.$$

**Definition 2** *(Diameter of Domain): Given a function $f(\mathbf{w})$, where $\mathbf{w} \in \mathcal{W}$, and $\mathcal{W}$ is $f$'s domain of definition. The diameter of $\mathcal{W}$ is denoted by $\Gamma$: for every $\mathbf{w}_1, \mathbf{w}_2 \in \mathcal{W}$: $||\mathbf{w}_1 - \mathbf{w}_2|| \le \Gamma$.*

**Theorem 1** *(Guaranteed Convergence Rate): If a convex function $f(\mathbf{w})$ is L-Lipschitz continuous function, then $||\nabla f(\mathbf{w})|| \le L$. Let $\Gamma$ be the diameter of domain. Applying equations (3)(4)(5) for gradients aggregation, we have the following convergence rate for the proposed FeDEC algorithm:*

$$f(\bar{\mathbf{w}}^T) - \min_{\mathbf{w} \in \mathcal{W}} f(\mathbf{w}) \le \mathcal{O}(\frac{\Gamma^2}{2\beta T} + \frac{\beta}{2}L^2), \tag{11}$$

*where $\bar{\mathbf{w}}^T$ is the average result of $\mathbf{w}$ for total training epoch $T$, $\beta$ is the learning rate in equation-(3), and $T$ is the total training epoch. If we let $\beta = \frac{\Gamma}{L\sqrt{T}}$, the convergence rate is $\mathcal{O}(\frac{1}{\sqrt{T}})$.*

**Proof**: To simplify the analysis, we consider fixed learning rate $\beta$. The proof includes the following steps:

(1) According to the definition of convex function,

$$f(\mathbf{w}^t) - f(\mathbf{w}) \quad \le \quad \nabla f(\mathbf{w}^t)(\mathbf{w}^t - \mathbf{w}).$$

(2) We define $G(\mathbf{w}) = \nabla f^T(\mathbf{w}^t)(\mathbf{w}^t - \mathbf{w})$, and $\mathbf{g}^t = \sum_{k=1}^{K} \pi_k \mathbf{g}_k^t$. The intermediate result of $f(\mathbf{w})$ in update time $t + 1$ is denoted by $\mathbf{w}_*^{t+1}$. With assumption 1 and assumption 2, we have:

$$
\begin{aligned}
G(\mathbf{w}) &= \frac{1}{\beta}(\mathbf{w}^t - \mathbf{w}_*^{t+1})(\mathbf{w}^t - \mathbf{w}) \\[2mm]
&= \frac{1}{\beta}((\mathbf{w}^t)^2 - \mathbf{w}^t\mathbf{w} - \mathbf{w}_*^{t+1}\mathbf{w}^t + \mathbf{w}_*^{t+1}\mathbf{w}) \\[2mm]
&= \frac{1}{2\beta}((\mathbf{w}^t)^2 - \mathbf{w}^t\mathbf{w} + \mathbf{w}^2 - (\mathbf{w}_*^{t+1})^2 + 2\mathbf{w}\mathbf{w}_*^{t+1} - \mathbf{w}^2 + (\mathbf{w}^t)^2 - 2\mathbf{w}^t\mathbf{w}_*^{t+1} + (\mathbf{w}_*^{t+1})^2) \\[2mm]
&= \frac{1}{2\beta}(||\mathbf{w}^t - \mathbf{w}||^2 - ||\mathbf{w}_*^{t+1} - \mathbf{w}||^2 + ||\mathbf{w}^t - \mathbf{w}_*^{t+1}||^2) \\[2mm]
&= \frac{1}{2\beta}(||\mathbf{w}^t - \mathbf{w}||^2 - ||\mathbf{w}_*^{t+1} - \mathbf{w}||^2) + \frac{\beta}{2}||\nabla f(\mathbf{w}^t)||^2 \\[2mm]
&= \frac{1}{2\beta}(||\mathbf{w}^t - \mathbf{w}||^2 - ||\mathbf{w}_*^{t+1} - \mathbf{w}||^2) + \frac{\beta}{2}||\mathbf{g}^t||^2.
\end{aligned}
$$

So we have:

$$
f(\mathbf{w}^t) - f(\mathbf{w}) \le \frac{1}{2\beta}(||\mathbf{w}^t - \mathbf{w}||^2 - ||\mathbf{w}_*^{t+1} - \mathbf{w}||^2) + \frac{\beta}{2}||\mathbf{g}^t||^2.
$$

(3) We project $\mathbf{w}_*^{t+1}$ to $\mathbf{w}^{t+1}$. With definition 1 and using non-expandable property of projection operation of convex set, we have:

$$
G_1(\mathbf{w}) \le \frac{1}{2\beta}(||\mathbf{w}^t - \mathbf{w}||^2 - ||\mathbf{w}^{t+1} - \mathbf{w}||^2) + \frac{\beta}{2}||\mathbf{g}^t||^2
$$

Due to L-Lipschitz Continuity (Lemma 1), we have:

$$
G(\mathbf{w}) \le \frac{1}{2\beta}(||\mathbf{w}^t - \mathbf{w}||^2 - ||\mathbf{w}^{t+1} - \mathbf{w}||^2) + \frac{\beta}{2}L^2
$$

So we have:

$$
f(\mathbf{w}^t) - f(\mathbf{w}) \le \frac{1}{2\beta}(||\mathbf{w}^t - \mathbf{w}||^2 - ||\mathbf{w}^{t+1} - \mathbf{w}||^2) + \frac{\beta}{2}L^2
$$

(4) According to definition 2, summing up all $\mathbf{w}$ from $t = 1$ to $T$, we have:

$$
\begin{aligned}
\sum_{t=1}^{T} f(\mathbf{w}^t) - Tf(\mathbf{w}) &\le \frac{1}{2\beta}(||\mathbf{w}^1 - \mathbf{w}||^2 - ||\mathbf{w}^{t+1} - \mathbf{w}||^2) + \frac{\beta}{2}L^2 T \\[2mm]
&\le \frac{1}{2\beta}|\mathbf{w}^1 - \mathbf{w}||^2 + \frac{\beta}{2}L^2 T \\[2mm]
&\le \frac{1}{2\beta}\Gamma^2 + \frac{\beta}{2}L^2 T.
\end{aligned}
$$

(5) According to Jensen's Inequality (Lemma 2), we have:

$$
\begin{aligned}
f(\bar{\mathbf{w}}^T) - f(\mathbf{w}) &= f(\frac{1}{T}\sum_{t=1}^{T}\mathbf{w}^t) - f(\mathbf{w}) \\[2mm]
&\le \frac{1}{T}\sum_{t=1}^{T} f(\mathbf{w}^t) - f(\mathbf{w}) \\[2mm]
&\le \frac{\Gamma^2}{2\beta T} + \frac{\beta}{2}L^2.
\end{aligned}
$$

We can get the result:

$$
f(\bar{\mathbf{w}}^T) - \min_{\mathbf{w}\in\mathcal{W}} f(\mathbf{w}) \le \mathcal{O}(\frac{\Gamma^2}{2\beta T} + \frac{\beta}{2}L^2),
$$

(6) Taking $\beta = \Gamma/(L\sqrt{T})$, the right part of the above equation becomes

$$\frac{\Gamma^2}{2\beta T} + \frac{\beta}{2}L^2 = \frac{\Gamma^2 L\sqrt{T}}{2\Gamma T} + \frac{\Gamma}{2L\sqrt{T}}L^2 = \frac{\Gamma L}{\sqrt{T}}.$$

Therefore we can obtain the simplified expression of the convergence bound $\mathcal{O}(\frac{1}{\sqrt{T}})$.

## A.2   EXPLANATION OF UNBIAS PARAMETER AGGREGATION IN SECTION 3.2.2

We compute the expectation of the aggregated parameters $\mathbf{W}_{mean}^t$ and $\mathbf{W}_{var}^t$ in Section 3.2.2 as follows.

$$
\begin{aligned}
\mathbb{E}[\mathbf{W}_{mean}^t] &= \mathbb{E}\left[\sum_{k=1}^K \pi_k \mathbf{W}_{mean}^{t,k}\right] \\
&= \sum_{k=1}^K \pi_k \mathbb{E}\left[\mathbf{W}_{mean}^{t,k}\right]
\end{aligned}
$$

$$
\begin{aligned}
\mathbb{E}[\mathbf{W}_{var}^t] &= \mathbb{E}\left[\frac{1}{||\mathbf{X}|| - K}\sum_{k=1}^K (||\mathbf{x}_k|| - 1)\pi_k \mathbf{W}_{var}^{t,k}\right] \\
&= \frac{1}{||\mathbf{X}|| - K}\mathbb{E}\left[\sum_{k=1}^K (||\mathbf{x}_k|| - 1)\pi_k \mathbf{W}_{var}^{t,k}\right] \\
&= \frac{1}{||\mathbf{X}|| - K}\sum_{k=1}^K (||\mathbf{x}_k|| - 1)\pi_k \mathbb{E}\left[\mathbf{W}_{var}^{t,k}\right]
\end{aligned}
$$

According to the above equations, if the parameters of the local models $\mathbf{W}_{mean}^{t,k}$ and $\mathbf{W}_{var}^{t,k}$ are unbias, then the aggregated model parameters are unbias as well.

### A.3 Structure of the Neural Network Models in Section 5

Here we report the detailed model structure used in the experiments. We use LeNet shown in Table 2 and the 4-layer CNN model shown in Table 3. We adopt a slim ResNet18 as shown in Table 4, where "Conv2d" is convolution layer, "BatchNorm2d" is batch normalization layer, and "Linear" is fully-connected layer. We can observe that every convolution layer is followed by a batch normalization (BN) layer. For all models, we use ReLU layer after every Conv2d layer. The structure of DenseNet121[2] and MobileNetV2[3] can be found in GitHub.

For language model, we consider the sentiment analysis task on tweets from Sentiment140 with 2-layer BiLSTM. The BiLSTM binary classifier containing 256 hidden units with pretrained 100-dimentional GloVe embedding. Each twitter account corresponds to a device.

Table 2: Detailed information of the LeNet architecture.

| Parameter | Shape | Layer hyper-parameter |
|---|---|---|
| Conv2d | $1 \times 16 \times 5 \times 5$ | stride=1, padding=2 |
| **BatchNorm2d** | $16 \times 2$ | - |
| MaxPool2d | - | stride=2, kernel size=2 |
| Conv2d | $16 \times 16 \times 5 \times 5$ | stride=1, padding=2 |
| **BatchNorm2d** | $16 \times 2$ | - |
| Conv2d | $16 \times 32 \times 5 \times 5$ | stride=1, padding=2 |
| **BatchNorm2d** | $32 \times 2$ | - |
| MaxPool2d | - | stride=2, kernel size=2 |
| Conv2d | $32 \times 32 \times 5 \times 5$ | stride=1, padding=2 |
| **BatchNorm2d** | $32 \times 2$ | - |
| Linear | $1568 \times 512$ | bias=True |
| Linear | $512 \times 128$ | bias=True |
| Linear | $128 \times 10$ | bias=True |

Table 3: Detailed information of the 4-L CNN architecture.

| Parameter | Shape | Layer hyper-parameter |
|---|---|---|
| Conv2d | $3 \times 32 \times 5 \times 5$ | - |
| **BatchNorm2d** | $32 \times 2$ | - |
| Conv2d | $32 \times 32 \times 5 \times 5$ | - |
| **BatchNorm2d** | $32 \times 2$ | - |
| MaxPool2d | - | stride=1, kernel size=2 |
| Conv2d | $32 \times 64 \times 3 \times 3$ | - |
| **BatchNorm2d** | $64 \times 2$ | - |
| Conv2d | $64 \times 64 \times 5 \times 5$ | - |
| **BatchNorm2d** | $64 \times 2$ | - |
| MaxPool2d | - | stride=1, kernel size=2 |
| Linear | $1024 \times 128$ | bias=True |
| Linear | $128 \times 10$ | bias=True |

---

[2]https://github.com/kuangliu/pytorch-cifar/blob/master/models/densenet.py
[3]https://github.com/kuangliu/pytorch-cifar/blob/master/models/mobilenetv2.py

Table 4: Detailed information of the ResNet18 architecture.

| Parameter | Shape | Layer hyper-parameter |
|---|---|---|
| Conv2d | $3 \times 64 \times 3 \times 3$ | stride=1, padding=1 |
| **BatchNorm2d** | $64 \times 2$ | - |
| Conv2d | $64 \times 64 \times 3 \times 3$ | stride=1, padding=1 |
| **BatchNorm2d** | $64 \times 2$ | - |
| Conv2d | $64 \times 64 \times 3 \times 3$ | stride=1, padding=1 |
| **BatchNorm2d** | $64 \times 2$ | - |
| Conv2d | $64 \times 64 \times 3 \times 3$ | stride=1, padding=1 |
| **BatchNorm2d** | $64 \times 2$ | - |
| Conv2d | $64 \times 64 \times 3 \times 3$ | stride=1, padding=1 |
| **BatchNorm2d** | $64 \times 2$ | - |
| Conv2d | $64 \times 128 \times 3 \times 3$ | stride=2, padding=1 |
| **BatchNorm2d** | $128 \times 2$ | - |
| Conv2d | $128 \times 128 \times 3 \times 3$ | stride=1, padding=1 |
| **BatchNorm2d** | $128 \times 2$ | - |
| Conv2d | $128 \times 128 \times 3 \times 3$ | stride=1, padding=1 |
| **BatchNorm2d** | $128 \times 2$ | - |
| Conv2d | $128 \times 128 \times 3 \times 3$ | stride=1, padding=1 |
| **BatchNorm2d** | $128 \times 2$ | - |
| Conv2d | $128 \times 256 \times 3 \times 3$ | stride=2, padding=1 |
| **BatchNorm2d** | $256 \times 2$ | - |
| Conv2d | $256 \times 256 \times 3 \times 3$ | stride=1, padding=1 |
| **BatchNorm2d** | $256 \times 2$ | - |
| Conv2d | $256 \times 256 \times 3 \times 3$ | stride=1, padding=1 |
| **BatchNorm2d** | $256 \times 2$ | - |
| Conv2d | $256 \times 256 \times 3 \times 3$ | stride=1, padding=1 |
| **BatchNorm2d** | $256 \times 2$ | - |
| Conv2d | $256 \times 512 \times 3 \times 3$ | stride=2, padding=1 |
| **BatchNorm2d** | $512 \times 2$ | - |
| Conv2d | $512 \times 512 \times 3 \times 3$ | stride=1, padding=1 |
| **BatchNorm2d** | $512 \times 2$ | - |
| Conv2d | $512 \times 512 \times 3 \times 3$ | stride=1, padding=1 |
| **BatchNorm2d** | $512 \times 2$ | - |
| Conv2d | $512 \times 512 \times 3 \times 3$ | stride=1, padding=1 |
| **BatchNorm2d** | $512 \times 2$ | - |
| Linear | $512 \times 10$ | bias = True |

## A.4 CONVERGENCE OF FEDERATED LEARNING ALGORITHMS FOR DIFFERENT MODELS ON DIFFERENT DATASETS

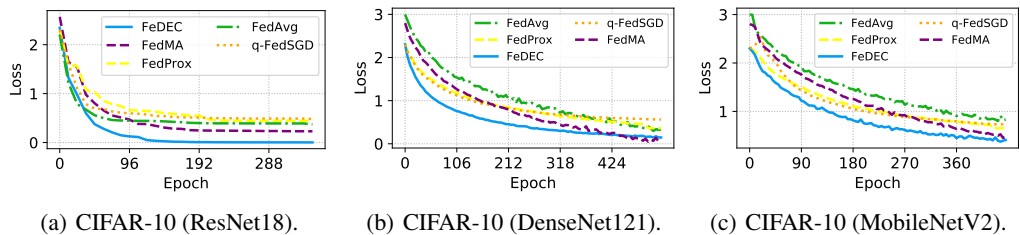

(a) CIFAR-10 (ResNet18).  (b) CIFAR-10 (DenseNet121).  (c) CIFAR-10 (MobileNetV2).

Figure 9: Convergence of different algorithms for different CNN models on CIFAR-10.

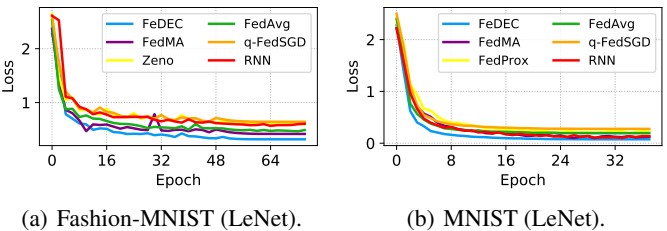

(a) Fashion-MNIST (LeNet).  (b) MNIST (LeNet).

Figure 10: Convergence of different algorithms for LeNet on FMNIST and MNIST.

## A.5 TRAINING EFFICIENCY OF FEDERATED LEARNING ALGORITHMS FOR DIFFERENT MODELS ON DIFFERENT DATASETS

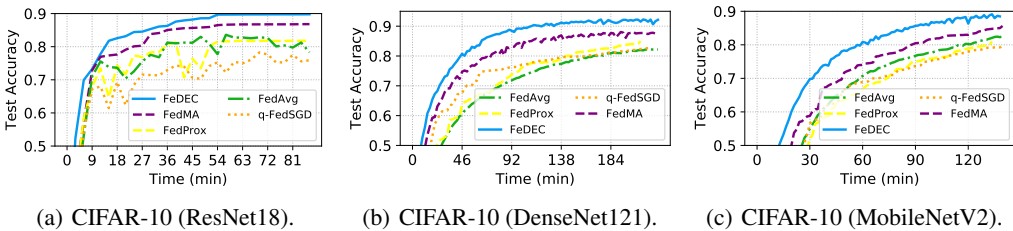

(a) CIFAR-10 (ResNet18).    (b) CIFAR-10 (DenseNet121).    (c) CIFAR-10 (MobileNetV2).

Figure 11: Training accuracy of different algorithms for different CNN models on CIFAR-10.

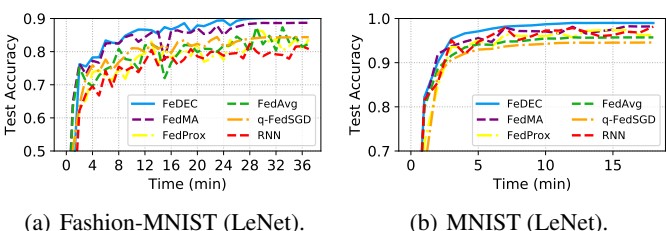

(a) Fashion-MNIST (LeNet).    (b) MNIST (LeNet).

Figure 12: Training accuracy of different algorithms for LeNet on FMNIST and MNIST.

---

**Algorithm 1:** FeDEC Aggregation

---

1 **Server**
2 **for** $t= 1\ to\ T$ **do**
3     Transmit $\mathbf{W}^t$ to all clients.
4     Receive $\mathbf{g}^t, \hat{\boldsymbol{\mu}}, \hat{\boldsymbol{\sigma}}$ from all clients.
5     Inference $\pi_k^t$ with 9 based on $\hat{\boldsymbol{\mu}}, \hat{\boldsymbol{\sigma}}$ from $\mathbf{W}_{mean}^t, \mathbf{W}_{var}^t$.
6     Update $\mathbf{W}_{NN}^{t-1}$ with 3 to get $\mathbf{W}_{NN}^t$.
7     Aggregate $\mathbf{W}_{mean}^t$ with 4 and $\mathbf{W}_{var}^t$ with 5.
8     Combine $\mathbf{W}_{NN}^t, \mathbf{W}_{mean}^t$ and $\mathbf{W}_{var}^t$ to new model $\mathbf{W}^t$.
9 **Client**
10 **for** $t= 1\ to\ T$ **do**
11     Receive server model $\mathbf{W}^t$.
12     Train local model $\mathbf{W}^t$ based on local dataset and get local gradients $\mathbf{g}^t$.
13     Transmit $\mathbf{g}^t$ and $\hat{\boldsymbol{\mu}}, \hat{\boldsymbol{\sigma}}$ to server.
14     Stop training until received server model $\mathbf{W}^t$.

---

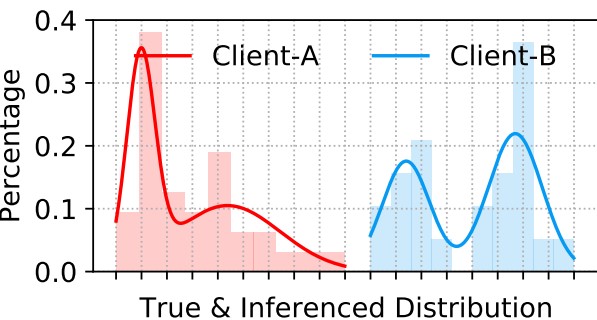

Figure 13: Variational inferenced approximate distribution of two clients of CIFAR-10 dataset experiment.

