# OpenReview forum: "Federated Learning with Decoupled Probabilistic-Weighted Gradient Aggregation"
_ICLR.cc/2021/Conference — Reject_

### Official Review · AnonReviewer4 · 2020-10-19
**Interesting ideas but many issues remain**

**Rating:** 3
**Confidence:** 3

**Review:**

In this work, the authors propose new ways of averaging updates received at the server from a subset of clients in a federated scenario. Specifically the authors aim to address issues arising from the non-iid nature of the data that arises in FL and propose to treat BatchNorm parameters differently from other NN parameters.

Introduction
Reading this paper, I am confused about terminology used by the authors. Specifically, the authors discuss 'data bias' and 'parameter bias'.
In in the introduction, the authors claim that 'Conventional approaches average gradients uniformly from the clients, which could cause great bias to the real data distribution'. Assuming the authors understand FedAvg to be a conventional approach, the averaging with $|x_k|/|x|$ (Eq. 1) is not uniform but weighted by the local dataset size. Further, the meaning of 'causing bias to the real data distribution' is not clear to me. Unfortunately, the further explanation and Figure 1 don't help me in understanding what is meant. The 'GroundTruth' distribution of labels for cifar10 is approximately uniform. From the context, I understand that the authors try to describe some consequence of a non-i.i.d sampled distribution of data according to labels, but I cannot understand the point they try to make.
Next, the authors discuss 'parameter bias'. The authors distinguish between BN parameters and other NN parameters. They term the BN parameters 'statistical parameters such as mean and variance'. Generally speaking, BN contains the 'scale and shift' parameters $\gamma$ and $\beta$ (https://arxiv.org/pdf/1502.03167.pdf), which I am assuming the authors reference to. Again, the authors make use of the term 'bias' to say: '[...] bias on the BN layer parameters'. I am not familiar with the notion of 'bias on a parameter' and would like the authors to clarify. Based on Figure 2 I assume they aim to convey that BN parameters in a FL setting converge to different values compared to a centrally trained model.
In Section 3.1 the authors further make the distinction explicitly between 'gradient parameters', by which they mean weights and biases as opposed to 'statistical parameters' of BN. Since the scale-and-shift parameters of BN are also updated by gradient descent, I am wondering if the authors mean the mean and variance estimate across data-points for feature-maps, which also plays a central part in (federated) BatchNormalization. The authors in their experimental section make no mention of how they form these global estimates for mean and variance in BN, so they omit that crucial detail there.

The authors specifically focus on label-skew as source of non-i.i.d-ness in this work, but never make this limitation explicit. Since the non-i.i.d challenges in FL are not limited to label skew, I believe the authors should make this explicit.

Related work:
The issues with BN in the federated setting has been described for example in Section 5 of https://arxiv.org/pdf/1910.00189.pdf. There, the authors propose to replace BN with GroupNorm, an approach that has been adopted in several follow-up and recent works in FL with models that originally contain BN. I would encourage the authors to compare their work against this approach, both in the RW section and also in the experimental section.

Method Section.
Notation-wise, I encourage the authors to not use $x$ or $x_k$ to denote a (labeled) dataset, since $x$ is usually reserved for a single data-point with associated label $y$. In Eq. (1) the loss formulation of FL is a bit sloppy since the parameters to optimise for, $W$ do not appear in the RHS of the equation. In FedAvg, we explicitly optimise $min \sum_k |x_k|/|x| L_k(W,x_k)$, where the local parameter estimates $W_k$ appear as intermediate parameters as a consequence of multiple local optimisation steps.
The authors claim that 'data points available locally could be biased from the overall distribution'. Again, I believe to understand the intended meaning to be the non-iid issue, but I encourage the authors to make their understanding of 'bias' more concrete.

In FedAvg, the individual clients do not transmit gradients $\nabla L_k ()$ to the server (Section 3.2.1). This is the approach in conventional distributed SGD as employed in a high-speed-connected data-centre for speeding up centralised training. In this centralised setting, the non-iid problem does not exist. Instead, in FL, clients transmit parameters that have been updated through a series of gradient-descent steps. More recent work (https://arxiv.org/abs/2003.00295) makes the role that these transmitted parameters have in an interpretation as a gradient. This distinction is important.

The derivation in Appendix A.2 for show-casing the unbiased-ness of the variance parameter averaging seems wrong. Going from the third to fourth equation makes a mistake and also if the original expectation was equal to the sum of weighted expectations, then simply averaging would actually be the unbiased estimator. The authors here are falsifying their own argument through a derivation mistake. The last equation should only pull the expectation into the sum in the right-most term and you are done.

Section 3.2.3
I like the notion of modelling the datasets as GMM and to infer responsibilities at averaging time. The explanation of the approach is confusing to me, however.
If I understand it right, then EQ 9 describes a VAE setup per client k. There is no sharing of parameters or latent space between clients. $s_k = [\mu_k,\sigma_k]$ are the mean and standard-deviation across the whole local dataset at a client $k$. The authors propose to encode this single vector $s_k$ into a latent space z_k of dimension $C$. The authors do not explicitly specify the prior p(z_k), but given the constraints on $z_k$, I assume it is meant to be a Dirichlet distribution. From context I could imagine that it has something to do with the per-client label distribution.
Since each ELBO is per-client and each client has just one data-point $s_k$, I do not understand the need for auto-encoding. Simply infer z given a decoder-model for the single data-point.  The authors mention the use of neural networks, but they do not detail their architecture choices anywhere. It is also unclear to me how $\pi_k$ falls out in equation 10. I imagine it I corresponds to some sort of posterior across all local models' encoding z. Since the latent-spaces across clients k do not share any meaning, I don't see how that can be sensible.  All together, this section is not readable to me.
I can see the appeal of reweighing updates through a specific formulation pi_k, but since the updates are label-independent (eq. 3,4,5), how does that play into this.

Section 4
My understanding of proofs of this form is somewhat limited. From what I can gather, this proof shows that FedAvg converges if the per-client loss-function is pre-multiplied by a constant factor $\pi_k$. As such, the convergence proof should be analogous to what is presented in e.g. https://arxiv.org/pdf/1907.02189.pdf with the exception of the update in Eq 5. Maybe the other reviewers can comment further on this.

Section 5.
In the Federated Setting, 20 clients should be considered not enough for experimental validation generally speaking. I appreciate the breath of experiments in terms of models, algorithms and related algorithms.
Unfortunately, the authors chose to define their own non-iid-split of the datasets. In general, I would appreciate to see comparisons with existing data-set splits in related work on non-iid data, such as for example https://arxiv.org/abs/2003.00295. This would help avoid the bifurcation of the literature.
The reference to q-FedSGD seems to be wrong.

After reading the experiment section, some serious questions arise:
What is the fraction of selected clients per communication round? What is the number of local epochs per client? Some of the related works seem to suggest performing FedSGD (also the first paragraph in Section 3.2.1 suggests this). If you have single-gradients per device and equal-size data-sets per client, then I don't see how non-iid data-distribution across clients are an issue as this approaches a global mini batch step, where each mini-batch consists of smaller mini-batches, one from each client. The non-i.i.d issue in FL stems from the fact that each client optimises on its own for a sufficiently long time that the resulting progress is destroyed by averaging in parameter space. The authors need to specify their setup here. Concretely, I would want to at least see Cifar10 split into 100 clients, 10 of which are selected at every round. Each client needs to optimise locally for a full epoch on its own dataset of 45000/100 = 450 data-points.

The authors propose two things: A new averaging approach by computation of $\pi_k$, as well as a new approach to estimating the gradient for the scale-parameter of BN using pooled averaging. These two things need to be studied separately by setting $\pi_k = |x_k|/|x|$ in one scenario. At the moment, my trust into the computation of $\pi_k$ is very low, since the corresponding section is not understandable to me. The pooled averaging approach seems sensible and I am curious to see if it solves the issue of BN in FL. Additionally, the authors need to clarify how the BN-statistics are computed at test time. I would also like to see a comparison of the proposed models with BN and with the state-of-the-art method which is replacing them with GroupNormalization.

Finally, I would like to thank the authors for the interesting approach to training BN-equiped Neural Networks in a federated setting. This idea seems promising to me. The approach for computing pi_k is not clear to me and I would like the authors to revise it. Please specify the role that knowledge of the label-distribution plays and if the method is applicable when non-iid-ness stems from other sources than label skew. (I propose looking at the FEMNIST dataset for example).
Many issues with this paper remain and I encourage the authors to overhaul their work.

I see no issues with the Code of Ethics

---

> ### Author Response · Authors · 2020-11-17
> **Response to Reviewer#4**
>
> We thank the reviewer for the detailed comments about this paper. And we hope that the reviewer first read "Response to all reviewers" which would explain part of the reviewer's concerns we think.
>
> [Motivation \& Terminology] Please reference "Response to all reviewers", we clarify our motivation again, and we hope it will be helpful. And for the BN parameter, what we want to express is only mean and variance in BN layer. We have update the new revision which contains an algorithm box to explain the experimental details.
>
> [Non-IID Settings] In Hyperparameter Analysis section, we study the settings both on label imbalance and dataset size imbalance, which typically are two major non-IID settings for federated learning. Maybe label skew is the area for byzantine problem in federated learning we think.
>
> [Related Work] We will compare our work with https://arxiv.org/pdf/1910.00189.pdf in the following version.
>
> [Derivation of Variance] We are really thankful that the reviewer point out this flaw in our paper, and we edit
> it in the new revision.
>
> [Auto-encoding of Mean and Variance] The global estimates for mean and variance is based on Eq.9. We can learn from [1] that after every Conv. layer, the feature maps are almost retains the main information of the images. And in [2] we find that the mean and variance of feature maps is also contains information which makes the network have a higher dependence of the input. So we want to embed the information of every-layer feature maps with variational auto-encoder to an latent space which has the distribution of feature map distribution. For every client, we only have one $\mathbf{s}_k$, but with all the clients we would have a distribution about $\mathbf{s}$ and the embedding space would also have a distribution. So when we sample from this distribution based on the priori of mean and variance we receive it from client-$k$ we can get a approximate distribution about the data distribution of client-$k$.
>
> [About $\pi_k$] In Eq.10, we assume that latent space sampled output is approximate distribution of data distribution at client-$k$. And we normalize the final $\pi_k$ based on the same class distribution among all $K$ clients first and then normalize it among all the different class. For example, if a class named \textit{bird}, we normalize the distribution weight among all $K$ \textit{bird} distribution weights and then sum all the normalized classes based on all kind of normalized distribution.
>
> [Section 4] Because we use decoupled method, so the model is updated in two parts: learned parameters part and statistical parameters part. In this section, we prove that the convergence rate about learned parameters because it is under a learning process. As for Eq.4 \& Eq.5, it is statistical average with $\pi_k$ and we state the unbias-ness in Appendix A.2. Because
> mean and variance do not show in chain rules of differential，it's better to not include these parameters
> into gradient descent proof section.
>
> [Section 5] We should admit that 20 is not a large number of cilents in federated learning. But as same of [3],
> using 20 clients is enough to convey the correctness and solidness with large accuracy gap between pre-works and the proposed method. And we are truely willing to consider the data partition method in https://arxiv.org/abs/2003.00295. And we modify the reference about q-FedSGD. For the experimental details, we select all the clients every communication round,  local epochs equal to 1. The data-set size is not equal from client to client, and we do not do the gradient step for mini batch step, we do it after a full epoch training process, this is what we study at. And the Cifar10 splited into 100 clients with 45000/100 = 450 data-points which
> is equal data-set size. Our focus is how to get high accuracy when class imbalance and data-set size imbalance exist in the same time, spliting datas into 20 client have shown the large heterogeneity in this settings, and the results have been
> shown in Figure.7 \& Figure.8. So multiple clients experiments can be added here, but it is not our focus and it is not necessary in this scene.
>
> [Comparison with GroupNormalization] In GN paper, Figure.5 shows the GroupNormalization replacement results. We can see that top-1 validation accuracy with BatchNorm and GroupNorm for LeNet over CIFAR-10 with 5 partitions are 79.7\% and 70.0\% for IID and Non-IID settings with FedAvg algorithm. And our algorithm with LeNet (with BN) in CIFAR-10 top-1 accuracy are 78.23\% and 75.41\% for IID and Non-IID settings. And the result shows the robustness with data imbalance and dataset size imbalance.
>
>
> [1] https://arxiv.org/abs/1311.2901
>
> [2] https://arxiv.org/abs/1806.02375
>
> [3] https://arxiv.org/abs/2002.06440

---

> > ### Comment · AnonReviewer4 · 2020-11-24
> > **Thank you for the rebuttal**
> >
> > I have read through the other reviewer's comments as well as you general answer and answer to my review. I have gone through the updated paper again and tried to approach it with a fresh mind. Unfortunately, I still do not completely understand what is happening. Notation is confusing and changing throughout. The intuition I develop from this submission is that there are valuable insights and novel contributions - generative modelling of feature-mean and variance is a really interesting idea. Also the experimental results look really promising.
> > I encourage the authors to re-write their paper with a focus on notation consistency and on clarity of writing. Especially since the concepts of 'mean and variance parameters', 'BN parameters' and bias of different things that are used in this paper are easily to confuse (as is the case for me), I believe a proper definition of these terms - following established notation - would really increase readability.
> > I encourage making the distinction between FedSGD (single update per client) and FedAvg (>1 update) - at this moment, the authors discuss FedSGD but experiment with FedAvg across different sections of the paper.
> >
> > In light of my confusion, I will reduce my confidence, but I will stick to my evaluation of the paper.
> > I am looking forward to reading a revised version of this work to properly understand the ideas behind it!

---

> > > ### Author Response · Authors · 2020-11-25
> > > **Thank you for response**
> > >
> > > We replace all the 'BN parameters' expression with 'means and variances in BN layers', which we distinguish $\mu, \sigma$ with $\gamma, \beta$, and we hope that will make it clear to all the reviewers. And we have rewrote the whole notations in entire section-3 and section-4, and we make all notations consistent in these sections. But $\mathbf{s}^t\_k$ is a notation of $\mathbf{W}^{t,k}\_{mean}$, $\mathbf{W}^{t,k}\_{var}$ for convenient and we do not change it. But in section-4, the convergence analysis is a general case for FeDEC algorithm, we can not just replace all $\mathbf{w}$ to $\mathbf{W}^t_{NN}$ in order not to lose generality.
> > >
> > > We clearly notice the difference between FedSGD and FedAvg. The baseline we discuss FedAvg after one full local training with whole local data. And the FeDEC is also transmit pseudo-gradients, which are modified in section-3.2.1, after one full local training with whole local data. In the contrary, as you said, FedSGD can transmit gradients with only small batch local training that will lead to no non-IID settings. We think this is the biggest difference between FedSGD and FedAvg, so we think the baselines we discussed is fairly right.

---

### Official Review · AnonReviewer2 · 2020-10-20
**A novel and effective model aggregation method for federated learning**

**Rating:** 6
**Confidence:** 4

**Review:**

##########################################################################

Summary:

The paper proposed FedDEC, a novel approach to conduct model updates aggregation in federated learning. The main motivation of this paper is to decouple the aggregation of normal model weights and statistics in BNs separately such that both data and model heterogeneity can be handled. Theoretical analysis indicates that the proposed FedDEC method enjoys a good convergence guarantee. Extensive experimental results are provided to show that FedDEC enjoys high efficiency and better model accuracy under the non-IID environment compared to the considered baseline methods.

##########################################################################

Reasons for score:

Overall, I think the current manuscript is marginally above the acceptance threshold of the ICLR conference. Studying the effectiveness of the model aggregation process in federated learning is a promising research direction. The proposed approach to conducting the decoupled model aggregation over the model parameters and statistics aggregated in the BN layers is promising. The proposed FedDEC method is justified both theoretically and empirically. Theoretically, the authors show that FedDEC enjoys the same convergence rate as normal SGD. Extensive experimental results over the simulated federated learning environment indicate that FedDEC enjoys good training effectiveness while reaching to better model accuracy. If my concerns on the current manuscript (please see "Cons") are addressed, I will be happy to improve my evaluation score.

##########################################################################

Pros:

1. The paper is well written. The research direction on improving the model aggregation in federated learning is promising and practical.

2. The formulation and theoretical analysis of the proposed FedDEC method looks promising.

3. Extensive experimental results are provided for the image classification tasks under the simulated non-iid environment, which indicates that FedDEC enjoys high effectiveness in improving the model test accuracy under the data heterogeneity.


##########################################################################


Cons:

1. A detailed discussion of the algorithmic aspect of FedDEC is missing. Section 3 discusses the formulation of FedDEC, however, it’s not super clear on the exact steps that FedDEC will run e.g. what information will the clients upload to the data center?; what are the exact operations that the data center will conduct? The authors’ are highly encouraged to add an algorithm box to describe the FedDEC algorithm.
2. From the description in Section 3, it seems FedDEC always assumes the local clients to train only ONE local epoch. If that’s the case, then the communication efficiency of FedDEC can be worse than FedAvg due to higher communication frequency. A more detailed discussion on this can be helpful to understand FedDEC better.
3. The convergence rate of FedAvg has been explicitly studied in [1]. It would be helpful to make a clearer comparison between the convergence rates of FedDEC and FedAvg.
4. In Section 3.2.2., the mean and variance in BN are referred to as BN layer parameters. However, in a BN layer, there are two trainable weights (\gamma + \beta [2]) and mean + variance statistics collected over the training batches. It would be helpful to make the notations clearer.
5. Although language models usually do not have BN layers, it’s still easy to imagine that the FedDEC algorithm can be applied over language tasks. Adding a language processing task can make the current manuscript stronger.

[1] https://arxiv.org/pdf/1907.02189.pdf

[2] https://arxiv.org/pdf/1502.03167.pdf


#########################################################################


Minor Comments:

1. Typos in the draft: (i) in the abstract “suffer form” —> “suffer from“; (ii) “$W_{NN}$ (parameters of parameters of BN layers)” —> “$W_{BN}$ (parameters of parameters of BN layers)” in Section 3.2.
2. Missing references: [1-4].

[1] https://arxiv.org/pdf/1907.02189.pdf

[2] https://arxiv.org/pdf/1812.01097.pdf

[3] https://arxiv.org/pdf/1908.07873.pdf

[4] https://arxiv.org/pdf/1912.04977.pdf

---

> ### Author Response · Authors · 2020-11-17
> **Response to Reviewer #2**
>
> We thank the reviewer for their careful review of the paper, and we address the Cons part as following.
>
> [Algorithm Box] We should add a algorithm box in it, and in the new revision, we have updated it at the last page. We
> hope that will help understand our algorithm.
>
> [Communication Efficiency] We should admit that we have higher communication frequency than FedAvg algorithm. As we said in "Response to all reviewers", our focus is how to alleviate data bias and parameter bias so that we can reach a higher accuracy and faster convergence, and we made it. And we are willing to study communication frequency reducing problems during the following works. Thanks for indicating that.
>
> [Convergence Rate of FedAvg] We have noticed that [1] has explicitly studied the convergence rate of FedAvg. But model aggregation and gradient aggregation are totally different, and comparing the convergence rate in numbers would not shows the truely convergence speed. But in Figure.4 \& Figure.9 we can find that our algorithm can converge faster and descent deeper than FedAvg in real experiments.
>
> [BN Parameters] We have change the parameters in new revision, please confirm that we have modified it without confusion.
>
> [Language Models] We do the additional experiments on language models in Sentiment140 dataset with 2-layer BiLSTM. The BiLSTM binary classifier containing 256 hidden units with pretrained 100-dimentional GloVe embedding. Each twitter account corresponds to a device. And the result shows bellow (because LSTM do not have BN layers, so the FeDEC models ):
>
> Table 1. Sentiment140 binary classification results based on 2-layer BiLSTM.
>
> |    Dataset     | Central | FedAvg | RNN  | FedProx | q-FedSGD | FedMA  | FeDEC  |
> | :------------: | :-----: | :----: | :--: | :-----: | :------: | :----: | :----: |
> | BiLSTM@Sent140 | 81.47%  | 72.14% |  -   | 71.08%  |  68.44%  | 73.81% | 77.51% |
>
> Thank you again for reviewing.

---

### Official Review · AnonReviewer1 · 2020-10-27
**An interesting view on parameter aggregation for federated learning which devises specific parameter aggregation rules to fit the nature of different layer types.  The proposed solution and its fundamental idea are however a bit arbitrary (in many technical choices) and perhaps flawed. Clarifications and further elaborations are needed.**

**Rating:** 3
**Confidence:** 3

**Review:**

PAPER SUMMARY

This paper presents a new federated learning scheme for neural network that accounts simultaneously for different layer types and different (heterogeneous) local data distributions. In particular, the proposed method differentiates between parameter updates for conv, fc versus bn (batch-norm) layers.

The main argument for this differentiation is that aggregating parameter gradient for bn layers is vulnerable to high bias when local data are heterogeneously distributed. This leads to a new proposal on aggregating bn parameters directly (instead of using their gradient).

Both parameter gradient and parameter updates are weighted by the same set of probabilistic weights that seem to be associated with mixture weights of a GMM model that generate local training data. These weights are learned via a deep generative model that encode local statistics of local data into a latent space, which seems to be associated with random drawn from a (latent) categorical distribution associated with those mixture weights.

A theoretical convergence guarantee for the proposed algorithm is also provided.

NOVELTY & SIGNIFICANCE

This paper poses an interesting view on the parameter aggregation of federated learning. I appreciate the authors' perspective that perhaps the aggregation of parameters defining different layer types need to be diversified to fit the nature of the data.

However, I find the proposed solution and its fundamental idea of modeling such heterogenity somewhat arbitrary and perhaps flawed. To elaborate:

First, it is not at all clear to me why the authors associate the weights that combine parameter gradients (for fc and conv) and parameters (for bn) with the mixture weight that generate local data. This seems like an arbitrary choice to me and there seems to be a flaw here: if a local device receives very little data but its data come from a mixture component with large weight, its gradient will likely be biased (due to the lack of data) but will still dominate others (due to its large mixture weight) -- I would like to hear the authors' thoughts on this.

Second, if I understand correctly, the auto-encoder model devised to learn those mixture weights is supposed to encode the local statistics (i.e., mean & variance estimates of a Gaussian that fit the local training input) to a categorical distribution over label classes. I am not sure if I missed something here but this is a bit strange: by right, the mean and variance of the *input* distribution do not contain any information about its *label* -- how does the embedding manage to generate information that does not exist in the first place? Likewise, how would such *label* information be related to the mixture weights of the *input* distribution via Eq. (10) -- could the authors elaborate further on this?

Third, on a high-level of idea, I am also not fully convinced that the aggregating the bn parameter gradient is vulnerable to high bias. This statement somehow came across as a politically correct statement with no concrete substantiation. It will be better if there is some technical demonstration that explicitly show that for bn layer, the bias of the aggregated gradient is higher than the bias of its aggregated parameters.

Last, I find the theoretical analysis is not at all particular about the inner working of the proposed algorithm. The analysis is on one hand generally applicable to any function that satisfies the stated assumption and, on the other hand, not specific enough to account for the facts that the combination weights are also being updated and that the bn parameters are not updated via gradient aggregation. Given this, I do not think the analysis is applicable to this setting, and it should be evaluated separately in its own setting.

TECHNICAL SOUNDNESS

I have made high-level check over the main bulk of technical derivation and have not spot any glaring issues. But, as I said above, on the idea level, the three technical contributions here (separating bn aggregation from those of fc and conv, learning the mixture weights from local statistics (barring access to data) and the theoretical analysis) are orthogonal with little connection to one another -- for the first two contributions, there seems to be flaws on the idea level that need further clarification.

EXPERIMENT

The reported results appear positive but it seems the authors only generate those for one single run. As the improvement margin is relatively small, it is important to average results over multiple runs to make sure the improvement is significantly above the deviation margin.

Furthermore, reporting only the final performance gives very little insight regarding how the invented components help avoid accumulating high bias, and also whether the auto-encoding scheme correctly recover the mixture weight -- perhaps this can be shown by evaluating the proposed mechanism on a synthetic dataset where we know the ground-truth.

---

> ### Author Response · Authors · 2020-11-17
> **Response to Reviewer#1**
>
> We thank the reviewer for proposing careful comments about this paper.
>
> [Very Little Data Device] First, we should clarify that the local data was not generated by GMM. The data exists in client itself, and we assume that the multiple client datas was a mixture distribution, which is what we want to infer what the distribution is. So it not possible one client that data come from a mixture component with large weight. But it is possible one client has little data, we explain this scene. In Eq.10, we normalize the final $\pi_k$ based on the same class distribution among all $K$ clients first and then normalize it among all the different class. For example, if a class named *bird*, we normalize the distribution weight among all $K$ *bird* distribution weights and then sum all the normalized classes based on all kind of normalized distribution. So if a client has little data, it can not have large weights in every class. And if it has large weights in several classes, we can also normalize it through the second normalization because the distribution weights of other classes in this client is almost 0 and will cause integrally low aggregation weights.
>
> [Label Information in Eq.10] What we get to infer the approximate distribution of label distribution is the  statistical information of feature maps after every Conv. layer. We can learn from [1] that after every Conv. layer, the feature maps are almost retains the main information of the images. And in [2] we find that the mean and variance of feature maps is also contains information which makes the network have a higher dependence of the input. So we want to embed the information of every-layer feature maps with variational auto-encoder to an latent space which has the distribution of feature map distribution. So when we sample from this distribution based on the priori of mean and variance we can get a approximate distribution about the data distribution. Because the input information is a transformation of label distribution, so we just say it is an "approximation", but it show that the approximate distribution fits well to true distribution in the new revision page-20. We hope this explanation helps.
>
> [BN bias] For the third question, I think the reviewer want to express that the aggregation method is vulnerable to high bias and why aggregate gradient rather than parameter. We have demonstrate the unbias-ness of aggregation mean and variance in Appendix A.2. Aggregating gradient and aggregating parameter are two main stream in federated learning, but it is not related with BN bias. We aggregate BN layer parameter independently with the learning parameters, and the learned parameters just need to consider gradient/parameter aggregation. So whether we choose gradient aggregation or parameter aggregation, the BN aggregation method would not change.
>
> [Theoretical Results] Please reference what we state in "Response to Reviewer#3 [Theoretical results]...", thank you.
>
> [Technical Contributions] What we want to do is to alleviate the data bias and parameter bias so that we can get higher accuracy during federated learning. We find that if we do not decouple the parameters (as shown in Table 1 which called FeDEC(w/o)), only
> using inferred weights would not alleviate parameter bias. So we add decoupled aggregation (as shown in Table 1 which called FeDEC) to alleviate parameter bias and it shows that combining inferred weights and decoupled method makes it possible to get higher accuracy and faster convergence speed in federated learning. From our point of view, they are not orthogonal to each other.
>
> [Experiments] We have update the experiments information in the new revision, including average results, error bars in main pages and inferred distribution/true label distribution comparison in page-20. And the accuracy improvement is 7\% compared with FedAvg and 3\% compared with SOTA FedMA, it is really a large improvement. Furthermore, the biases are independent during each step and we will evaluate it after each step, so there is no accumulated bias.
>
> Thank you again for reviewing.
>
> [1] https://arxiv.org/abs/1311.2901
>
> [2] https://arxiv.org/abs/1806.02375

---

### Official Review · AnonReviewer3 · 2020-10-28
**Not good enough**

**Rating:** 4
**Confidence:** 4

**Review:**

1/ Summary

This paper introduces an aggregation mechanism designed for neural networks with batch normalisation layers. This mechanism relies on two parts: probabilistic mixing weights of the loss function and the use of a weighted pool estimator for aggregating the BN variance parameters. The mixing weights are derived from a GMM with variational inference. A convergence result in the *convex* case is provided. Experimental results on 3 image datasets show that this approach yields better results than other standard FL algorithms (FedAvg, FedProx, q-FedSGD, FedMA…) as well as a better resilience to heterogeneity (understood as class imbalance).

2/ Acceptance decision

Despite its seemingly good experimental results, I am in favour of rejecting this paper as correcting its weak points would require major changes.

3/ Supporting arguments

A/ The theoretical results (Sec 4, one page) are irrelevant to the problem tackled by the paper.

Indeed, they are proved in the case of a convex function. However, the proposed method is relevant for neural networks with batch normalisation layers, so more than 1 layer, and therefore these networks yield non-convex loss functions, which do not satisfy the hypothesis of the results. Further, the unbiased gradients assumption (assumption 1) seems dubious in the non-iid case tackled by the authors. Last, but not least, it is difficult to understand if this assumption is valid or not (and the same holds for the results) because the notations used to state the theorem in Eq 11 do not correspond to the notations of the loss function Eq 2.

B/ Important technical details are omitted, which makes it difficult to understand the whole algorithm, threatens reproducibility, and shades some doubts on the experimental results.

- In section 3.2.3, the authors introduce a GMM for the mixing weights \pi_k while dropping the upper script t for convenience, which is understandable. However, it is not stated how the estimation process of \pi_k^t evolves throughout the different batches. Does one need to do 1 full variational inference at each aggregation step? Can one re-use previous parameters through aggregation rounds? Does one rely on constant \pi_k?
- A related question is the behaviour of the proposed method when reducing communication frequency. Indeed, the algorithm is only exposed in the case of communication after each local gradient computation, which is typically avoided in the edge setting considered by the authors. How is it extended in this case? In particular, how does the estimation of the parameters \pi_k change?
- In a related note, the number of local computations done before any aggregation is omitted in the experimental section. Are all models compared with the same number of local steps?

C/ Writing is not very clear and should be improved. In particular, some notations do not make sense; notations tend to change a lot across the different sections, making it difficult to see a global picture emerge.

4/ Additional comments

- Delving into appendix A.2 which explains the aggregation rule for variance layers, I do not understand the transition from the 3rd equation to the 4th (from top to bottom). Could the authors elaborate more on it?
- Although the approach is « decoupled », the mixing weights used in the aggregation rules (Eq 3) (standard layers) and ( Eq 4) (BN means) are the same. Have the authors tried completely decoupling these mixing weights?
- In Section 3.2.3, if I understood correctly the means \mu_k and \sigma_k are related to the empirical means and variances of the batches in the different clients. Sharing means and variances of each batch could yield to large privacy leaks. This remark goes together with the communication frequency remark above.
- In Section 3.2.3, the authors state that they assume that data are drawn from a Gaussian distribution « without loss of generality ». However, it is well known that e.g. image distributions are not Gaussian.
- In Section 5.1, I do not understand the notation $pr(x) = \mathcal{N}(0, 1)$. How can a probability take negative values?
- How does performance of the different methods vary with respect to the communication frequency? This is an important metric of the training efficiency, and missing from the experiments.
- Given that the experiments have been run multiple times, it would be nice to provide error bars to understand the significance of the different results.
- Some typos:
    - page 4,  $W_{BN}$ instead of $W_{NN}$ at the beginning of Sec 3.2
    - "Form" -> "from" in the introduction
- The presentation of figures 4, 5, 6 make it difficult to read the captions.

---

> ### Author Response · Authors · 2020-11-17
> **Response to Reviewer #3**
>
> We greatly appreciate the reviewer’s detailed review and suggestions to improve the paper, not only for technical details but also for writings.
>
> [Theoretical results]
>
> For the recently published papers like [1, 2] etc. are all proved that the convergence guarantees of proposed algorithms in convex functions, and prove algorithm can converge in convex function also gives us a theoretical guidance to apply algorithms to non-convex functions. And the (Assumption 1) is not related to non-IID settings, because it just reflect that the gradient we receive at server from client-$i$ is unbiased about the gradient descent amplitude of $f(\cdot)$ in client-$i$, it has no information about other clients and so it is not dubious. Eq.11 we shows the general case about our algorithm  and the $\mathbf{w}$ related to $\mathbf{W}_{NN}$, we must state it clearly and thanks for indicating it.
>
> [Technical Details]
>
> For the $\pi_k$ and $\pi^t_k$, $\pi_k$ is used for convenience when we infer it during epoch-$t$. The model aggregation and variational inference process are executed alternately, because they all need converge with gradient descent method. So we need to make a inference process in server at each aggregation step.
>
> We avoid the communication frequency reducing because we need statistical parameters changes to make  good variational inference, so we make the local step equals to 1, so as the baselines. And as we state in "Response to all reviewers", our focus is how to alleviate data bias and parameter bias so that we can reach a higher accuracy and faster convergence, and we made it. And we are willing to study communication frequency reducing problems during the following works.
>
> [Additional]
>
> - The variance expection should be $\frac{1}{||\mathbf{X}|| - K} \sum_{k=1}^K (|| \mathbf{x}_k|| - 1) \pi_k \mathbb{E}(\mathbf{W}^{t,k}_{var})$. Thanks for indicating it.
>
> - The standard layers is updated with weighted SGD, and the BN means is updated with weighted average.
>
> - The $\mu_k$ and $\sigma_k$ is get from models layers which are not the true data distribution parameter mean and variance, so it will not cause large privacy leaks. Just because it is not the true data distribution, we use variational inference to infer the approximate distribution parameters.
>
> - The $\mathcal{N}(0,1)$ is just a notation we sample the samples with a gaussian distribution probability. We are sorry to make confusion and we change it into $\mathcal{N}(0.5,1)$ without confusion.
>
> - We provide the error bars of BN parameters of Figure.6 and accuracy analysis of Figure.7/8 in the new revision with edits highlighted in red color.
>
> Thank you again for reviewing.
>
> [1] https://arxiv.org/abs/1907.02189 (ICLR'2020)
>
> [2] https://arxiv.org/abs/1902.00146 (ICML'2019)

---

### Author Response · Authors · 2020-11-16
**Response to all reviewers**

We thank all reviewers for their time and detailed comments about this paper. Most of the comments are helpful to improve this paper. Based on the problems we focus, we first address shared concerns and then respond to specific comments for every reviewer. And we have updated the paper with edits highlighted in red color. And we replace all the 'BN parameters' expression with 'means and variances in BN layers', which we distinguish $\mu, \sigma$ with $\gamma, \beta$, we hope that will make it clear to all the reviewers. And we have rewrote the whole notations in entire section-3 and section-4, and we make all notations consistent in these sections. But $\mathbf{s}^t\_k$ is a notation of $\mathbf{W}^{t,k}\_{mean}$, $\mathbf{W}^{t,k}\_{var}$ for convenient and we do not change it. But in section-4, the convergence analysis is a general case for FeDEC algorithm, we can not just replace all $\mathbf{w}$ to $\mathbf{W}^t_{NN}$ in order not to lose generality.

[Basic Problems]

In this paper, we focus on the accuracy improvement and convergence speed in federated learning when data bias and parameter bias in Non-IID incurs.

(I). If data is $\mathbf{x}_i$ in client-$i$, the learning model can be seen as fitting function $f(\mathbf{x}_i)$ of data, so the averaging method (with/wothout $|\mathbf{x}_i|/|\mathbf{x}|$) in FedAvg for models can be seen as  another type of averaging datas. This is making great data selection bias for different labels. As shown in Figure.1, using the  FedAvg method in Non-IID environment is just like sampling datas uniformly that the sampled data distribution can not fit the true distribution of all cilents.

(II). Due to the data is Non-IID for every cilent, so the model parameters is also biased. And the statistical parameters are truely reflection about data samples, the statistical parameters are all biased for all clients.

[Contributions]

In this paper, we first notice the bias problems in federated learning. We proposed FeDEC, a novel objective that decouple model aggregation in federated learning and infer the approximate distribution of different client, which shows that we can infer the approximate distribution about datas from the only information we get from models without privacy leakage and with the distribution information we can aggregate models in a decoupled way to alleviate parameter bias.

This is the first work we are aware of to explore such an objective in distributed/federated machine learning. Decoupled model aggregation enables the learned parameters $\mathbf{W}_{NN}, \gamma, \beta$ been learned with SGD, and statistical parameters $\mu, \sigma$ been updated with inferred weights to alleviate parameter bias. We proposed variational auto-encoder to encode the statistical parameters of feature maps into latent space, which is distribution of distribution. The feature map distribution can be applied as approximation of data distribution. If we get approximate data distribution, we can alleviate data bias. Besides, we theoretically prove that the convergence is guaranteed using our inferred weights for SGD, and we prove that less bias of statistical parameters. Our empirical evaluation on a set of real-world datasets and deep learning models demonstrates the higher accuracy we can get.

[Methods]
- [Decouple Aggregation] We did experiments in tensorflow and pytorch and find that when transmitting models in these two frameworks they would not transmit $\mu, \sigma$ of BN layer. But if you transmit gradients, all the detailed parameters will be counted. The decouple method is just detach the parameters learned with SGD, which includes Conv. filters and $\gamma, \beta$ parameters in BN layer. We update these parameters with a distributed gradient descent way. As for statistical parameters, we use weighted average method for $\mu$ and weighted pooled variance method for $\sigma$. This modification makes it possible to communicate statistical parameters to adjust activation of every Conv. layers and get a higher accuracy improvement.
- [Variational Inference] We can only get model information from all the client without privacy leakage. BN layer after every Conv. layer shows the mean and variance of datas feature maps, which can be seen as a non-linear transformation of all the data. So the bunch of these parameters is another type of information about data distribution, we infer the approximate distribution of datas from these information for every client and it will direct us to have a better aggregation in server.

As we state at the beginning, we focus on the accuracy improvement and convergence speed when alleviating the both bias on data and parameter. In order to address the important contributions, we did not focus on the communication cost and multi-local steps training, but our fast convergence speed shown in Figure.4 indicates that to reach fixed accuracy, FeDEC needs fewer communication steps. But we are willing to add the additional experiments about it.

---

### Decision · Program_Chairs · 2021-01-07
**Final Decision**

**Decision:**

Reject

**Comment:**

The authors’ feedback has not fully addressed the reviewers’ concerns and the reviewers think that the paper is not ready for the publication. The authors should consider the following issues for the future submission:

1) The concern from Reviewer 1: if a local device receives very little data but its data come from a mixture component with large weight, its gradient will likely be biased (due to the lack of data) but will still dominate others (due to its large mixture weight).

2) Numerical experiments are not consistent with theoretical results. The theory is for convex but experiments are with non-convex loss. The response from authors does not resolve this issue.

3) Notation is confusing and changing throughout. We strongly suggest the authors revise carefully this and make it clear.

Although the experimental results are potential, we would like the authors to revise it carefully by addressing the reviewers’ concerns and further improve it by considering theoretical results for non-convex in order to submit to the next venues.